# Modelling and Kinetic Study of Novel and Sustainable Microwave-Assisted Dehydration of Sugarcane Juice

**Tayyaba Alvi** [1], **Muhammad Kashif Iqbal Khan** [2,*], **Abid Aslam Maan** [2], **Akmal Nazir** [3], **Muhammad Haseeb Ahmad** [4], **Muhammad Issa Khan** [1], **Muhammad Sharif** [5], **Azmat Ullah Khan** [6], **Muhammad Inam Afzal** [7], **Muhammad Umer** [7], **Shabbar Abbas** [7] and **Shahnah Qureshi** [1]

[1] National Institute of Food Science and Technology, University of Agriculture, Faisalabad 38000, Punjab, Pakistan; tayabaalvi@gmail.com (T.A.); drkhan@uaf.edu.pk (M.I.K.); shahnahqureshi@gmail.com (S.Q.)
[2] Department of Food Engineering, University of Agriculture, Faisalabad 38000, Punjab, Pakistan; abid.maan@uaf.edu.pk
[3] Department of Food, Nutrition and Health, College of Food and Agriculture, United Arab Emirates University, Al Ain 15551, UAE; akmal.nazir@uaeu.ac.ae
[4] Institute of Home and Food Sciences, Faculty of Life Sciences, Government College University, Faisalabad 38000, Punjab, Pakistan; haseeb1828@gmail.com
[5] Institute of Animal and Dairy Sciences, University of Agriculture, Faisalabad 38000, Punjab, Pakistan; drsharifuaf@yahoo.com
[6] Department of Food Science and Human Nutrition, University of Veterinary and Animal Sciences, Lahore 54000, Punjab, Pakistan; azmat.khan@uvas.edu.pk
[7] Department of Biosciences, COMSATS University, Islamabad, Park Road, Tarlai Kalan, Islamabad 45550, Pakistan; inam.afzal@comsats.edu.pk (M.I.A.); muhammadumer@comsats.edu.pk (M.U.); shabbar.abbas@comsats.edu.pk (S.A.)
* Correspondence: kashif.khan@uaf.edu.pk

**Abstract:** Sugarcane juice is a perishable food with a good nutritional profile. Thus, there is a need to increase its shelf life by reducing water content which facilitates storage and transportation. In this study, process conditions were optimized to concentrate the sugarcane juice at various microwave powers (30, 50, 80, 100 W). A central composite design was applied to optimize the process conditions (power and time). The overall evaporation time depends on microwave powers; increase in power reduced the processing time. The results showed that at 100 W sugarcane juice was concentrated to 75° brix for 15 min which reduced the energy consumption to 1.3 times compared to other powers. Moreover, microwave processing better retained the sensory properties of concentrate and preserved its antioxidant activity. Thus, 100 W was most energy efficient in concentrating sugarcane juice. In general, microwave processing reduced the processing time and cost making it a sustainable approach to concentrate juices.

**Keywords:** kinetics; modelling; sugarcane; microwave; dehydration; sustainability; brix

## 1. Introduction

*Saccharum officinarum* is commonly called sugarcane and is a major cash crop of Pakistan. The sugarcane juice is obtained by milling of sugarcane and the remaining fraction is obtained from milling of bagasse and straws. Both are burned in various industries as fuel source [1]. Sugarcane juice is comprised of 75–85% water, 10–21% non-reducing sugars (sucrose), 0.3–3% reducing sugars (fructose and glucose), 0.5–1% organic substances, 0.2–0.6% inorganic substances and 0.5–1% nitrogen

containing compounds [2]. The sugarcane is mainly grown for industrial production of table sugar (sucrose); however, a limited quantity of it is utilized for preparation of traditional brown sugars locally known as gurr and shaker. The concentration and clarification process during sugar manufacturing produces various by-products, i.e., molasses and bagasse. Some other applications of these by-products are in chemicals, plastics, synthetic fiber, animal feed and ethanol production [3].

Sugarcane juice contains various bioactive components that make it a healthy drink. These bioactive compounds strengthen the immune system. It also reduces the inflammation, respiratory infections, digestive disorders and liver diseases. However, it has very short life span due to microbial and enzymatic activities. Thus, there is a need to preserve the juice for its availability throughout the year and maintain the antioxidant activity [4].

Currently, some advanced techniques (including membranes separation, spray drying, etc.) are being investigated to replace conventional evaporation-crystallization process [5,6]. However, microwave has not been used for concentration of sugarcane juice. It rapidly heats up the materials and does not affect the bioactive components. Thus, this study focuses on a liquid concentrated product through microwave (MW). Although, the MW technique has been investigated to replace conventional dehydration of fruits and vegetables [7], to our knowledge it has not been studied so far for evaporation or concentration of sugarcane juice.

The main aim of the study is to evaluate modeling mass transfer kinetics of microwave dehydration using different models. In this regard, sugarcane juice was selected as model juice for evaporation by using microwaves. Moreover, concentrated product can be reconstituted as a drink or can be directly added into different recipes. It is expected that it would be beneficial compared to table sugar. Thus, there is a need to investigate how evaporation proceeds as a function of time and microwave power.

## 2. Materials and Methods

### 2.1. Sugarcane Juice Extraction

The sugarcane juice was extracted with a lab-scale roller crusher. After extraction, the juice was filtered through muslin cloth to separate straw and other impurities with subsequent further processing.

### 2.2. Concentration of Sugarcane Juice through Microwave

The concentration of juice was carried out in a microwave oven (Orient, Karachi, Pakistan) with a capacity of 23 L. The juice samples (200 mL) placed in pyrex beaker were heated at different microwave powers (30, 50, 80, 100 W) to achieve 75° Brix. The experimental conditions were optimized by using response surface methodology based on time and power. A $2 \times 2$ central composite design of response surface methodology was used to determine the effect of power and time on moisture ratio and evaporation rate of the sugarcane juice. The concentrates were packed in airtight glass jars and stored at 4 °C for two hours.

### 2.3. Evaporation Rate

During evaporation phase, the weight of samples was periodically recorded at an interval of 3 min, and this evaporation rate (ER) was calculated by using equation:

$$ER = \frac{W_t - W_{t+dt}}{dt} \tag{1}$$

where, ER is the evaporation rate expressed in g water/100 g.min and dt represents the evaporation time in minutes. While, $W_t$ and $W_{t+dt}$ are the moisture contents at time t and t+dt, respectively [1].

### 2.4. Moisture Ratio (MR)

Water removed from samples was calculated as a function of MR with the following equation:

$$MR = \frac{M_t - M_e}{M_o - M_e} \quad (2)$$

where, $M_t$ are the moisture contents at any time and $M_e$ are the initial moisture contents before concentration. While $M_e$ is moisture content at equilibrium [2]. The evaporation curves were compared with various models as described in Equations (3)–(11) [1,3,4].

$$\text{Midilli } MR = a \times \exp(-kt^n) + bt \quad (3)$$

$$\text{Page } MR = \exp(-kt^n) \quad (4)$$

$$\text{Henderson and Pabis } MR = \exp(-kt) \quad (5)$$

$$\text{Logarithmic } MR = a \times \exp(-kt) + c \quad (6)$$

$$\text{Two} - \text{term } MR = a \times \exp(-k_0 t + b \times \exp(-k_1 t)) \quad (7)$$

$$\text{Two term exponentials } MR = a \times \exp(-kt) + (1 - a) \times \exp(-kat) \quad (8)$$

$$\text{Wang and Sing } MR = 1 + at + bt^2 \quad (9)$$

$$\text{Diffusion approximation } MR = a \times \exp(-kt) + (1 - a) \times \exp(-kbt) \quad (10)$$

$$\text{Verma et al. } MR = a \times \exp(-kt) + (1 - a) \times \exp(-ct) \quad (11)$$

The experimental data was used to determine diffusivity coefficient by using Fick's law of diffusion. As dehydration process is in unsteady state diffusion process through infinite slab, thus Fick's law can be written as;

$$MR = \frac{M_t - M_e}{M_o - M_e} = \frac{8}{\pi^2} \exp\left(\frac{-\pi^2 Dt}{4L^2}\right) \quad (12)$$

where, D is the effective diffusion coefficient ($m^2$/s), and L is the half-thickness (m) of imaginary slab of juice. By plotting ln MR versus time, effective diffusion coefficient [4] can be calculated by determining the slope ($\alpha$). This may be defined with the help of Equation (12), as follows:

$$\alpha = \frac{-\pi^2 D}{4L^2} \quad (13)$$

The dependence of effective diffusion coefficient on microwave power was estimated through modified Arrhenius model [3] as follows:

$$D = D_f \exp\left(-\frac{E_a m}{P}\right) \quad (14)$$

where, $D_f$ is the pre-exponential factor of the Arrhenius equation ($m^2$/s), $E_a$ is the activation energy (W/kg), $m$ is the average of sample mass (kg) and $P$ is the microwave power output (W). Furthermore, Equation (14) was written in a logarithmic form and ln (D) was plotted as a function of ($m/P$). The slope of the plot represents the values of $E_a$ and $D_f$ [3].

### 2.5. Free radical Scavenging Activity (DPPH)

The concentrated juice samples were evaluated for their capacity to scavenge the free radical DPPH (2,2-diphenyl-2-picrylhydrazyl). The samples were centrifuged (Eppendorf, Hamurg, Germany) at 15 °C and 6000 rpm for 5 min. The supernatant layer (0.1 mL) diluted up to 0.5 mL by adding ethanol (70%). Furthermore, 0.5 mL of DPPH (2,2-diphenyl-2-picrylhydrazyl) in a concentration of 0.2 mM

was added and mixed using vortex mixer (IKA-Werke GmbH & Co. KG, Staufen, Germany). The final mixture was incubated for 15 min at room temperature in darkness. The absorbance of samples was determined by using spectrophotometer (IRMECO, Geesthacht, Germany) at 517 nm. The decrease in the absorbance was calculated by Equation (15) and expressed as percent free radical scavenging activity [5,6].

$$\text{Scavenging activity (\%)} = \frac{A_o - A_s}{A_o} \times 100 \tag{15}$$

where, $A_o$ and $A_s$ are the absorbance values of blank and sample, respectively.

## 2.6. Energy Consumption

Specific Energy consumption of the drying process was expressed in kJ/kg water evaporated. So specific energy consumption can be calculated as;

$$Specific\ energy\ consumption = \frac{3.6 \times E_m}{(M_o - M_t) \times m_s} \tag{16}$$

Microwave energy ($E_m$) to concentrate the sugarcane juice till 75° brix at various powers was calculated by equation:

$$E_m = p \times t \tag{17}$$

where, p represents the microwave power (W) and t is the time in seconds [1,7] and $m_s$ is the mass of dry matter (kg)

## 2.7. Solubility Index

For the determination of solubility index, 2.5 mL of concentrate was mixed in 100 mL of distilled water at room temperature. The mixtures were stirred for 1 min in vortex mixer (IKA-Werke GmbH & Co. KG, Staufen, Germany) and then placed at 37 °C in water bath for 30 min. Afterwards, mixture was centrifuged (Eppendorf, Hamberg, Germany) at 3500 rpm and 4 °C for 20 min. Thereafter, supernatant layer was separated and water was evaporated, at 105 °C [8]. Solubility index was calculated by equation:

$$Solubility\ Index\ (\%) = \frac{Dried\ supernatant\ Weight}{Initial\ Sample\ Weight} \times 100 \tag{18}$$

## 2.8. Color

The color of the concentrate was measured using the CIE-lab SPACE, (Color Tec-PCM, NY, USA). Color analysis was determined by the procedure described by (Kortei et al., 2015 and Khan et al., 2016) [9,10]. The sample was placed in the colorimeter and readings of L*, a* and b* were taken in triplicates. By using these parameters, change in color (ΔE), Chroma (C*) and whiteness index (WI) were determined by using Equations (19)–(21), respectively:

$$\Delta E = \sqrt{(\Delta L^*)^2 + (\Delta a^*)^2 + (\Delta b^*)^2} \tag{19}$$

$$C^* = \sqrt{(a^*)^2 + (b^*)^2} \tag{20}$$

$$WI = 100 - \sqrt{(100 - L^*)^2 + (a^*)^2 + (b^*)^2} \tag{21}$$

## 2.9. Statistical Analysis

All measurements were carried out in triplicates and results are expressed as means ± standard deviation. The statistix 10 (Analytical Software, Tallahassee, FL, USA) was used for data analysis.

The influence of time and power on MR and endoplasmic reticulum (ER) was determined analysis of variance and Tukey's test was used to determine difference between treatments. Besides, statistical tests were carried out that are described in the following equations:

$$\text{Root mean square error (RMSE)} = \sqrt{\frac{1}{N}\sum_{i=1}^{n}\left(V_{\text{exp},i} - V_{\text{model},i}\right)^2} \tag{22}$$

$$\text{Sum of square error (SSE)} = \frac{1}{N}\sum_{i=1}^{n}\left(V_{\text{exp},i} - V_{\text{model},i}\right)^2 \tag{23}$$

$$\text{Chi square (CS)} = \frac{1}{N - n_p}\sum_{i=1}^{n}\left(V_{\text{exp},i} - V_{\text{model},i}\right)^2 \tag{24}$$

$$\text{Relative percent deviation (RPD)} = \frac{100}{N}\sum_{i=1}^{n}\frac{\left|V_{\text{exp},i} - V_{\text{model},i}\right|}{V_{\text{exp},i}} \tag{25}$$

where, N represents the number of observations, $n_p$ indicates number of parameters, $v_{\text{exp},i}$ and $v_{\text{model},i}$ are the experimental and model values of i-th observation, respectively. Best model was selected based on coefficient of determination ($R^2$). Moreover, root mean square error, Chi square (CS) and relative percent deviation (RPD) indicates variation (goodness of fit) in model and experimental values. RMSE and CS values closer to zero indicate the closeness of the model to experimental data. Similarly, RPD determines the absolute difference between model and experimental values. RPD value less than 10% indicates that fit is good [11,12].

## 3. Results and Discussions

### 3.1. Evaporation Rate

The evaporation rate (Equation (1)) at different microwave powers is plotted as a function of time (Figure 1a). The results showed that evaporation rate was not constant throughout the process; initially the evaporation rate increased rapidly until the 50% percent moisture contents of the juice were removed. Afterwards, its trend started to reduce and resulted in a parabolic curve. Effect of power, time and moisture content were studied by applying central composite design for optimization of evaporation rate (Figure 1a–c). The surface plot showed that the evaporation rate increased with the increase in microwave powers (Figure 1a). When power increased from 30 to 100 W, evaporation rate increased from 3.72 to 12.70 g/100 g.min. This increase in evaporation rate at 100 W was about four times higher than 30 W. it resulted in the reduction of overall processing time to fifteen minutes. The increase in evaporation rate was due to higher heating rate at 100 W compared to 30 W of microwave power. Thus, more vapors were produced, resulting in higher evaporation rate.

Initially, higher evaporation was due to higher moisture content of sample. This was supported by plotting the evaporation rates as a function of moisture content (Figure 1b,c). It is obvious that the initial increase in evaporation rate is linked to high initial moisture contents of sugarcane juice that resulted in a greater microwave power absorption and higher evaporation rate. As the moisture content decreases in the sample, microwave power absorption also decreases, which results in decreased evaporation rate. Other authors also reported that when moisture content reduces there is a reduction in the evaporation rate [1].

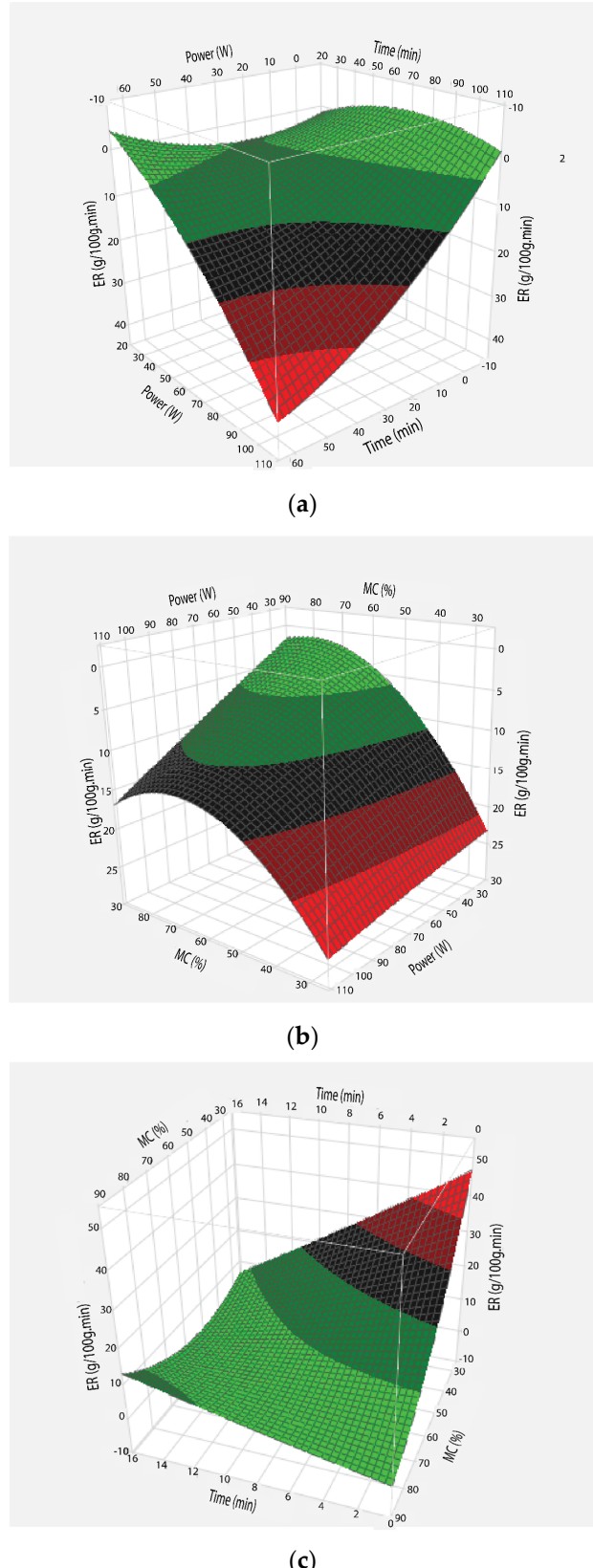

**Figure 1.** A graphical representation of evaporation rate influenced by the interaction of time, power and moisture contents. (**a**) Evaporation rate as a function of time and power; (**b**) Evaporation rate as a function of power and moisture content; (**c**) Evaporation rate as a function of time and moisture content.

### 3.2. Moisture Ratio

Moisture ratio at various powers was calculated by Equation (2) and results are shown in Figure 2. To visualize the combined effects of time and power on moisture ratio, the dehydration of sugarcane juice was studied by applying central composite design and the response surface graph was plotted (Figure 2a). For all microwave powers, the moisture ratio of samples decreased with the passage of time, but at variable rates. However, the moisture ratio curves varied significantly as a function of microwave powers. This may be attributed to the rapid heating at higher powers as discussed earlier. Thus, there was a rapid reduction in moisture ratio at high microwave power, which resulted in a steeper curve compared to evaporation at low power. However, experimental moisture ratios are compared with predicted moisture ratios, which indicates that both values are close to each other (Figure 2b).

Moreover, nine thin-layer models were compared with experimental moisture ratio and the model with higher $R^2$, and lower RMSE or $\chi^2$ values was considered as the best one. The drying kinetics of sugarcane juice was best described by Midilli model as supported by the statistical analysis; higher $R^2$ (0.998) and lower RMSE (0.014) and $\chi^2$ ($2.164 \times 10^{-4}$) values for various microwave powers (Table 1). Furthermore, the Midilli model values were compared with experimental moisture ratio (Figure 2c) and were in good agreement.

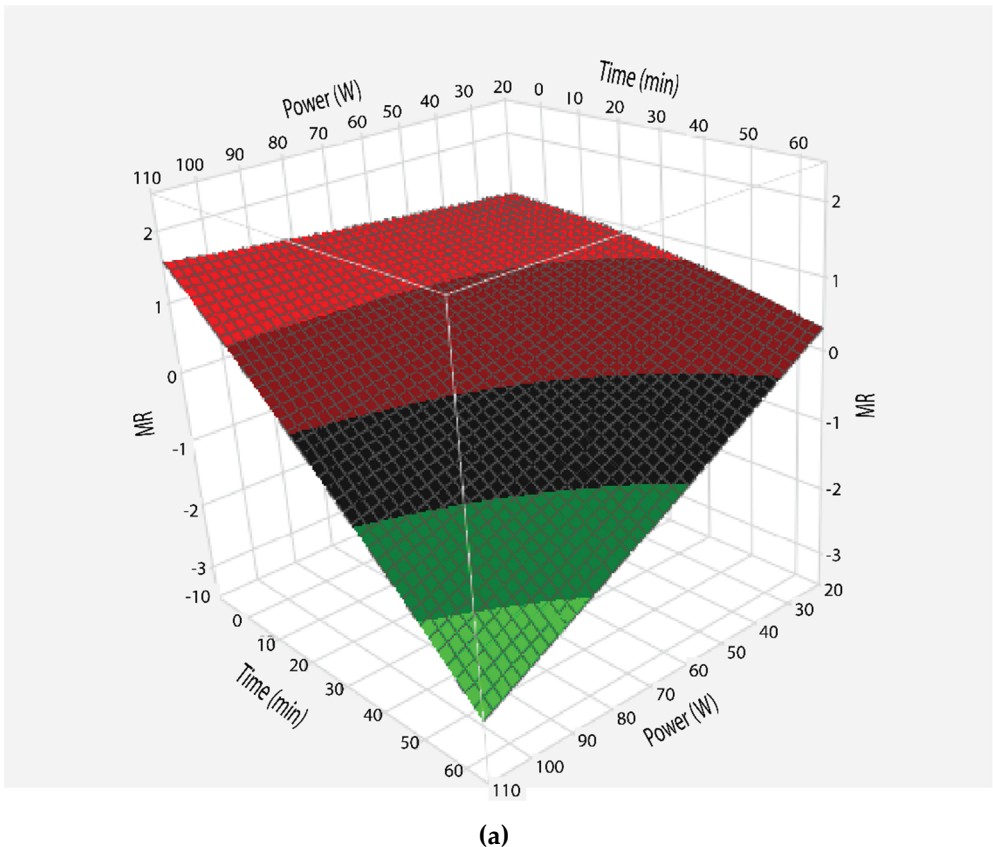

**(a)**

**Figure 2.** *Cont.*

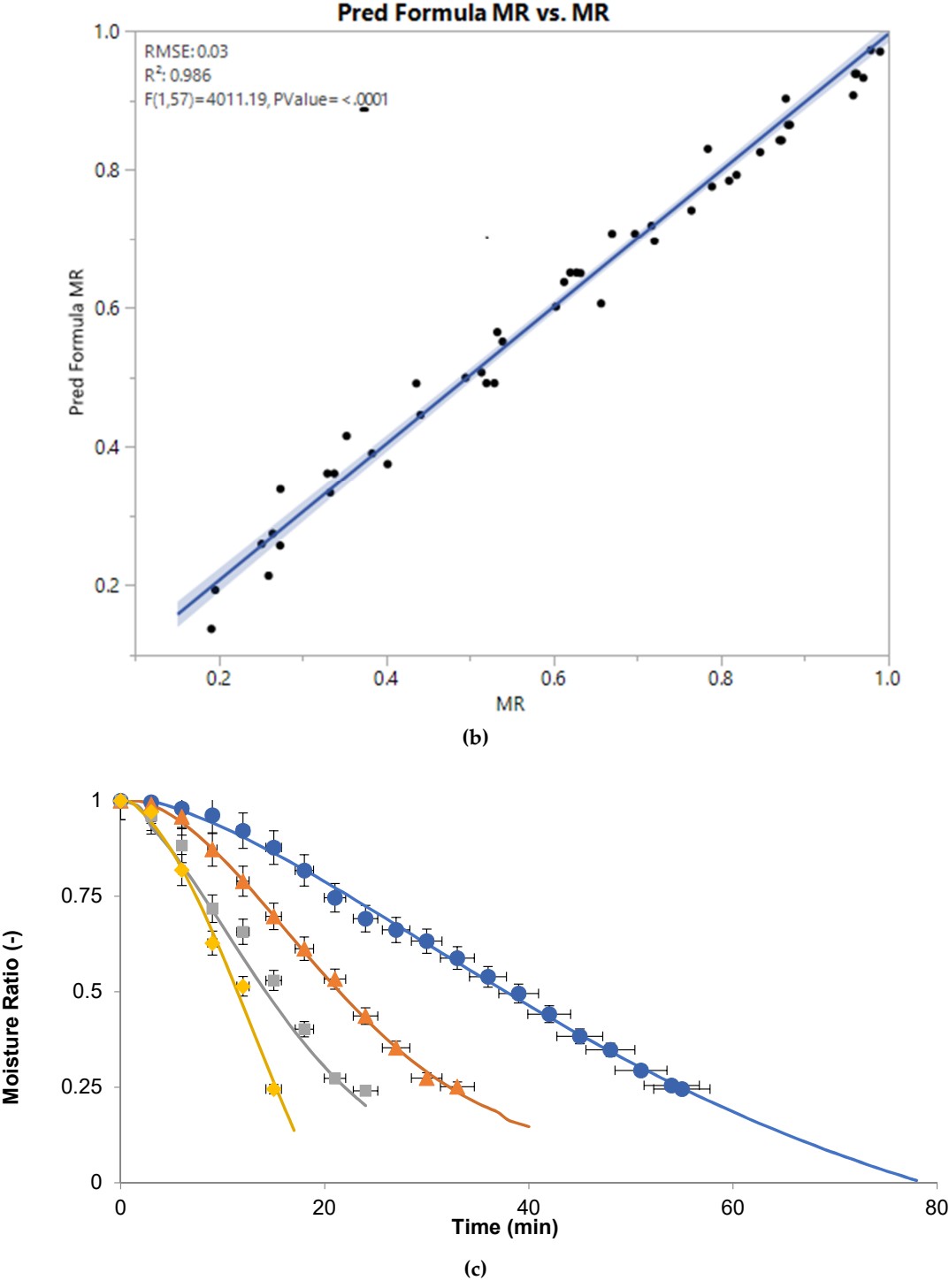

**Figure 2.** Response surface plot of moisture ratio (MR) as a function of time and power (**a**), comparison of experimental vs predicted MR (**b**), graphical representation of MR as a function of time at various microwave powers: (◊) 100 W, (□) 80 W, (△) 50 W, and (○) 30 W. Lines represent Midilli model for respective powers (**c**).

**Table 1.** A statistical evaluation of various kinetic models for microwave assisted evaporation at different powers.

| Model Name | Power | Model Constants | | | | | | | | $R^2$ | Chi ($\chi^2$) | SSE | RMSE | RPD (%) |
|---|---|---|---|---|---|---|---|---|---|---|---|---|---|---|
| | | n | k | a | b | c | $k_o$ | $k_1$ | g | | | | | |
| Midilli | 30 | 1.613 | 0.002 | 1.016 | −0.002 | - | - | - | - | 0.997 | $2.164 \times 10^{-4}$ | $2.291 \times 10^{-4}$ | 0.014 | 0.107 |
| | 50 | 1.866 | 0.003 | 1.004 | 0.002 | - | - | - | - | 0.998 | $1.053 \times 10^{-4}$ | $1.17 \times 10^{-4}$ | 0.010 | 0.072 |
| | 80 | 1.413 | 0.003 | 1.014 | −0.021 | - | - | - | - | 0.986 | 0.001 | 0.001 | 0.031 | 0.458 |
| | 100 | 2.194 | 0.002 | 1.013 | −0.017 | - | - | - | - | 0.989 | 0.001 | 0.002 | 0.030 | 0.712 |
| Page | 30 | 0.001 | 1.742 | - | - | - | - | - | - | 0.996 | 0.125 | $2.735 \times 10^{-4}$ | 0.344 | 69.566 |
| | 50 | 1.830 | 0.002 | - | - | - | - | - | - | 0.998 | $1.492 \times 10^{-4}$ | $1.658 \times 10^{-4}$ | 0.012 | 0.345 |
| | 80 | 1.711 | 0.006 | - | - | - | - | - | - | 0.994 | $5.383 \times 10^{-4}$ | $6.28 \times 10^{-4}$ | 0.022 | 0.220 |
| | 100 | 2.122 | 0.004 | - | - | - | - | - | - | 0.986 | 0.152 | 0.002 | 0.349 | 74.049 |
| Henderson and Pabis | 30 | - | 0.022 | 1.123 | - | - | - | - | - | 0.949 | 0.004 | 0.004 | 0.059 | 4.048 |
| | 50 | - | 0.037 | 1.121 | - | - | - | - | - | 0.939 | 0.005 | 0.006 | 0.069 | 4.906 |
| | 80 | - | 0.047 | 1.081 | - | - | - | - | - | 0.951 | 0.005 | 0.005 | 0.063 | 5.167 |
| | 100 | - | 0.067 | 1.094 | - | - | - | - | - | 0.887 | 0.012 | 0.016 | 0.098 | 8.620 |
| Logarithmic | 30 | - | $3.041 \times 10^{-6}$ | $4.881 \times 10^3$ | - | $4.88 \times 10^3$ | - | - | - | 0.992 | $5.526 \times 10^{-4}$ | $5.851 \times 10^{-4}$ | 0.023 | 0.365 |
| | 50 | - | $6.202 \times 10^{-6}$ | $4.125 \times 10^3$ | - | $4.125 \times 10^3$ | - | - | - | 0.987 | 0.001 | 0.001 | 0.032 | 0.417 |
| | 80 | - | | | - | | - | - | - | 0.987 | 0.001 | 0.001 | 0.032 | 0.922 |
| | 100 | - | $1.042 \times 10^{-5}$ | $4.878 \times 10^3$ | - | $-4.877 \times 10^3$ | - | - | - | 0.959 | 0.004 | 0.006 | 0.059 | 3.090 |
| Two term | 30 | - | - | 356.226 | −355.234 | - | 0.049 | 0.049 | - | 0.995 | $3.612 \times 10^{-4}$ | $3.825 \times 10^{-4}$ | 0.018 | 0.848 |
| | 50 | - | - | 0.707 | 0.414 | - | 0.037 | 0.037 | - | 0.938 | 0.005 | 0.006 | 0.069 | 4.906 |
| | 80 | - | - | 0.684 | 0.398 | - | 0.047 | 0.047 | - | 0.951 | 0.005 | 0.005 | 0.063 | 5.167 |
| | 100 | - | - | 0.689 | 0.406 | - | 0.067 | 0.067 | - | 0.887 | 0.012 | 0.016 | 0.098 | 8.620 |
| Two term exponentials | 30 | - | 0.035 | 2.097 | - | - | - | - | - | 0.993 | $4.403 \times 10^{-4}$ | $4.662 \times 10^{-4}$ | 0.020 | 1.205 |
| | 50 | - | $8.74 \times 10^6$ | $3.592 \times 10^{-9}$ | - | - | - | - | - | 0.898 | 0.009 | 0.010 | 0.089 | 3.970 |
| | 80 | - | 0.074 | 1.998 | - | - | - | - | - | 0.982 | 0.002 | 0.002 | 0.038 | 2.056 |
| | 100 | - | 0.124 | 2.207 | - | - | - | - | - | 0.975 | 0.003 | 0.003 | 0.045 | 3.723 |

**Table 1.** *Cont.*

| Model Name | Power | Model Constants | | | | | | | | $R^2$ | Chi ($\chi^2$) | SSE | RMSE | RPD (%) |
|---|---|---|---|---|---|---|---|---|---|---|---|---|---|---|
| | | n | k | a | b | c | $k_o$ | $k_1$ | g | | | | | |
| Wang and Sing | 30 | - | - | −0.009 | $−8.782 \times 10^{-5}$ | - | - | - | - | 0.990 | $6.814 \times 10^{-4}$ | $7.215 \times 10^{-4}$ | 0.025 | 0.760 |
| | 50 | - | - | −0.016 | $−2.641 \times 10^{-4}$ | - | - | - | - | 0.984 | 0.001 | 0.001 | 0.035 | 0.932 |
| | 80 | - | - | −0.023 | $−2.714 \times 10^{-4}$ | - | - | - | - | 0.989 | 0.073 | 0.001 | 0.030 | 0.116 |
| | 100 | - | - | −0.016 | −0.002 | - | - | - | - | 0.989 | 0.001 | 0.002 | 0.030 | 0.104 |
| Diffusion approximation | 30 | - | 0.018 | 0.655 | 1 | - | - | - | - | 0.907 | 0.007 | 0.007 | 0.080 | 3.340 |
| | 50 | - | 0.086 | −334.513 | 0.996 | - | - | - | - | 0.997 | $2.082 \times 10^{-4}$ | $2.313 \times 10^{-4}$ | 0.014 | 0.784 |
| | 80 | - | 0.099 | −201.879 | 0.995 | - | - | - | - | 0.983 | 0.002 | 0.002 | 0.037 | 1.889 |
| | 100 | - | 0.058 | 0.639 | 1 | - | - | - | - | 0.855 | 0.015 | 0.020 | 0.111 | 6.965 |
| Verma et al | 30 | - | 0.018 | 0.781 | - | - | - | - | 0.018 | 0.907 | 0.007 | 0.007 | 0.080 | 3.340 |
| | 50 | - | 0.031 | 0.399 | - | - | - | - | 0.031 | 0.898 | 0.009 | 0.010 | 0.089 | 3.970 |
| | 80 | - | 0.042 | 0.500 | - | - | - | - | 0.042 | 0.931 | 0.006 | 0.008 | 0.075 | 4.455 |
| | 100 | - | 0.058 | 0.500 | - | - | - | - | 0.058 | 0.855 | 0.015 | 0.020 | 0.111 | 6.965 |

### 3.3. Effective moisture Diffusivity as Function of Microwave Power

The values of moisture diffusivities (D) were calculated by plotting ln MR verses dehydration time (Figure 3). Slope of the straight line led to the calculation of D for all microwave powers (Table 2). The moisture diffusivities for microwave powers at 30, 50, 80, 100 W were in range between 0.684 and 6.62 ($10^{-15}$ m$^2$/s). The results indicated that values effective moisture diffusivities increased with the increase in microwave power. This increase in D can be attributed to higher microwave power that elevated the sample temperature, thereby increasing the water vaporization. Similarly, change in diffusivities are reported in literature [13]. Moreover, the activation energy and pre-exponential factor ($D_f$) was calculated from Equation (14) [3] and their values were $1.981 \times 10^{-3}$ W/kg and $4.32 \times 10^{-9}$ m$^2$/s, respectively.

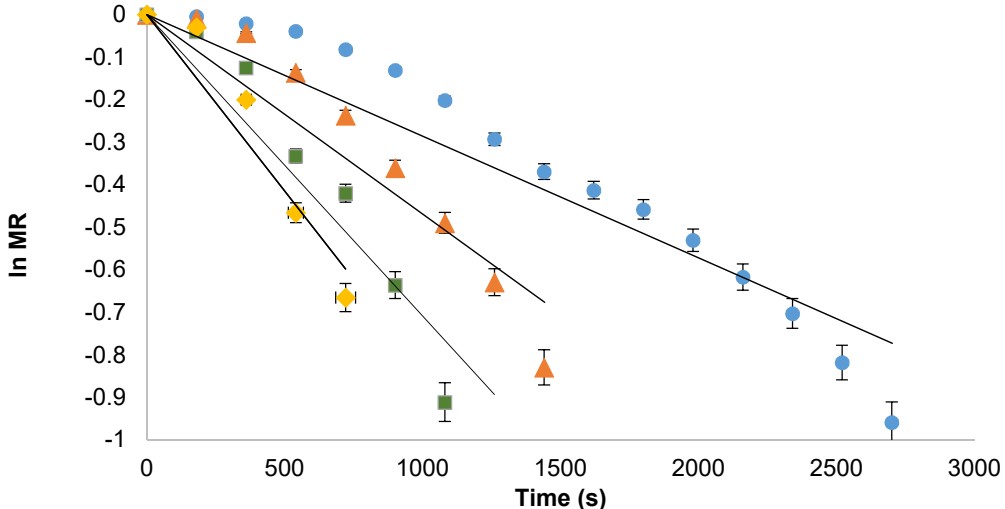

**Figure 3.** Graphical representation of ln MR as a function of time at various microwave powers; (◊) 100 W, (□) 80 W, (Δ) 50 W, and (○) 30 W.

**Table 2.** Effect of microwave power on effective diffusivity, energy consumption and antioxidant activity.

| Power W | Effective Diffusivity m$^2$/s ($\times 10^{-15}$) | Specific Energy Consumption kJ/kg | Antioxidant Activity % |
|---|---|---|---|
| **30** | 0.68 [d] | 452 ± 0.001 [b] | 72.98 ± 0.03 [a] |
| **50** | 1.45 [c] | 444 ± 0.01 [c] | 69.24 ± 0.05 [c] |
| **80** | 3.36 [b] | 555 ± 0.005 [a] | 70.54 ± 0.03 [b] |
| **100** | 6.61 [a] | 411 ± 0.002 [d] | 64.93 ± 0.03 [d] |
| ***p*-value** | <0.001 | 0.2049 | 0.0007 |

Mean values in columns carrying different letters are significantly different from each other.

### 3.4. Antioxidant Activity and Energy Consumption

The antioxidant activity of sugarcane concentrate made at different microwave powers is presented in Table 2. The effect of various microwave powers that exhibited that lowest power had highest scavenging activity compared to highest powers. The later one reduced the DPPH activity of concentrate. Scavenging activity increased from 64.93% to 72.98% when microwave power was decreased from 100 to 30 W, respectively. The statistical results indicate that all the results are significantly different from each other (*p* value: 0.0007). Arimandi et al. [14] stated that antioxidant activity reduced with the increase in power of microwave oven. The energy consumption to achieve 75° Brix of sugarcane juice was calculated, and outcomes are presented in (Table 2). The findings proved that 80 W consumes the highest energy compared to other ones but there is not much difference. While highest power (100 W) needed the lowest (411 kJ/kg) energy to achieve required brix. This may be attributed to lowest processing time (15 min) compared to others (25–55 min). Statistical findings also proved that energy

consumption at all powers is non-significant (*p* value: 0.2049). Thus, heating at 100 W is sustainable in terms of time, for the preparation of sugarcane juice concentrate. Hence, all the other parameters were analyzed at 100 W power.

*3.5. Solubility Index & Color Analysis*

Solubility index (SI) determines the reconstitution quality of the concentrate that is calculated through Equation (18) and results are illustrated in Table 3. The results indicated that SI was influenced non-significantly by the heating method. Conventional heating method exhibited lower SI value (45.28) compared to microwave. Similarly, microwave-based heating exhibited similar color properties compared to conventional ones (Table 3), i.e., ΔE values are non-significantly lower for microwave and C* values are non-significantly higher for microwave but as compared to fresh juice values are significantly different. Moreover, whiteness index for microwave was comparatively higher than conventional heating method.

**Table 3.** Color and solubility of sugarcane concentrate compared with fresh juice.

| Sample | ΔE | C* | WI | Solubility Index (%) |
|---|---|---|---|---|
| **Fresh juice** | 0 [b] | 21.13 ± 0.03 [a] | 28.90 ± 0.04 [ab] | N/A |
| **Conventional concentrate** | 18.01 ± 0.01 [a] | 3.85 ± 0.01 [b] | 27.06 ± 0.03 [b] | 45.28 ± 0.009 [b] |
| **Microwave concentrate** | 17.68 ± 0.04 [a] | 4.07 ± 0.08 [b] | 30.12 ± 0.005 [a] | 57.6 ± 0.01 [a] |

Means, within the columns, with different lettering are significantly different from each other ($p < 0.05$).

## 4. Conclusions

Sugarcane juice (as a model juice) was concentrated with microwave heating methods until 75° brix. The results indicated that evaporation rate and moisture ratio was highest at 100 W power of microwave. In comparison with other heating methods, microwave needs less time to concentrate the juice and exhibited higher evaporation rate and moisture ratio values. Similarly, the antioxidant activity of concentrate was influenced by the power of microwave, i.e., lowest power had the highest DPPH (2,2-diphenyl-2-picrylhydrazyl) values. In general, results exhibited that microwave processing is more sustainable in terms of time and energy consumption. Thus, the juices may be concentrated efficiently with microwave assisted technique.

**Author Contributions:** Supervision and conceptualization, M.K.I.K. and A.A.M.; Writing-original draft, T.A.; Statistical analysis, A.N., M.H.A. and M.I.K.; Conduction of experiments, T.A.; Review and editing, M.S., M.I.A., M.U.; Proofreading, A.U.K., S.A. and S.Q.

**Funding:** This research received no external funding.

**Acknowledgments:** Authors are thankful to Ali Hassan for his generous help in the design of experiments (DOE) of this research.

**Conflicts of Interest:** Authors declares no conflict of interest.

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
