# Peer review of "Modelling and Kinetic Study of Novel and Sustainable Microwave-Assisted Dehydration of Sugarcane Juice"

_processes, doi:10.3390/pr7100712_

Round 1

Reviewer 1 Report

Genaral in the text there are some errors e.g. line 82 - two dots, different font in tables and different in the text, Verma et al should be with dot: Verma et al.

line 21, 24 - 100 (W) without brackets

line 61 - how long material was storage at 4 °C?

line 65 - microwave power (30-100W) was applied continuously? It was a special microwave oven design for laboratory or used in homes?

line 65-66  - correct sentence starting from "The experimental conditions..."

line 107 - by who... please give names in the text.

line 116 - what type of sample? after treatment by microwave? it was juice? Please describe with details.

line 128 - if continuous then this calculation can be true

line 142, 146  - evaporation rate - once is written from small letter another time starts from Capital letter - correct in one style

figure 1, 3 - give the legend on a graph to be more clear

table 1 a - correct it - in horizontal page

line 196, 200 - all e  e.g. (e-15 m2/s) correct as x10-15 m2/s ect.

line 204 - first % is not necessary: 64.93 to 72.98%

line 208 - table 2 without brackets

table 2, 3 - all value are without standard deviation. Please add this and also mark homogeneous groups

line 217 - table 3 without dot

line 234 - delete this part of the sentence: "without affecting the nutraceutical protentional of bioactive compounds present in these juices". - it was not checked - you mean that DPPH measurement is enough?? It is not enough to give this type of statement, especially for microwave processes, which influence on the bioavailability of food compounds.

Reviewer 2 Report

Dear Authors,

Before publication, the manuscript should be prepared correctly. For me, in current form has serious flaws. You did not do simple statistical analysis, ex. ANOVA and then post-hoc tests to be sure if there are significant differences between values of various parameters. Without this analysis, your conclusions are not true. I recommend reading article ex. doi:10.3390/foods8010020. Moreover, Authors stated that "microwave has not been studied so far for evaporation or concentration of juices", but there is a lot of research concerning this problem. It is easy to find other publications where researchers used microwave to concentrate juices. (For example, J Food Sci Technol. 2016 May; 53(5): 2389–2395, doi: 10.1007/s13197-016-2213-0  or Article in Journal of Food Process Engineering 40(5):e12535 · October 2017, DOI: 10.1111/jfpe.12535

Below you can find what else should be corrected.

Lines 4-6 In affiliations, various size of font is used.

L30 "Saccharum officinarum" should be italic. Verb is missing.

L31 and 35/36 have the same information.

L32 and 39 - bagasse is a substrat and by-product?

L39 Dot is missing on the end of the sentence.

L47 and 50/51 - There are other publications where researchers used microwave to concentrate juices.

L55 - all bioactive compounds, but you had determined only DPPH/antioxidant activity. Where are determinations of other compounds?

L56 It was only lab-scale, so it is not easy to make industrial application using these same parameters and procedures. Other juices have different compounds, so different process parameters are needed and it is possible that results for other juices will be different.

L66 "based on were time and power" - unclear.

L68 how long samples were stored?

L74 100g.min?

L 79 coma after respctively.

L81 additional dot on the end of the sentence.

L104 Supernatant not supernatent

L108 - changes in color (dE) WERE determined. Eq. 16, 17 and 18 (not 12,14 and 15).

L122 - Eq. 19 not 15

L129 - Energy consumption is wrongly calculated, because there is not only power of microwave (W) x time (s). Authors in methodology cited publications where is this same wrong equation. Interestingly, in this second manuscript is reference to correctly calculated energy consumption (doi.org/10.1016/j.jssas.2012.09.002).

L130 - It was only goodness of fitting, but no an influence of time and microwave powers on various parameters.

L144 - where this is showed? I don not think that till 50% is linear increase....

L148 3.72 and 12.7 g - should be the same accuracy.

L165 "juice were studied by applying central composite" -unclear style

L166 - where are statistical data?

L176 - unclear style

L180 - dependence or dependent?

L179-185 this part is unclear for me. What those equations mean?

L190 - Wrong name of Table 1. For prediction of using some models should be name of Table similar to for ex. (doi:10.3390/foods8010020) The values of model constants, R2, Chi2, SSE, RMSE and RPD of modeling .... using Midilli, Page... model.

L191 - Page model should not be used for the prediction (for powers 30 and 100W) because RPD is too big. All the values should be the same accuracy!

For Wang and Sing model (80W) Chi2 is hudge - is not a mistake/wrong data?

Data of Table 1 should be more described in the manuscript.

L204 - "highest scavenging activity compared to highest ones" - should be rewritten.

L205 - "was decrease from 100 to 30 (W) are used" - style unclear

L208 - Table 2 should be without brackets.

L208 - "The findings proved that 80 (W) consume highest energy compared to other ones" - Why? How it could be explained?

L209-212 - It is not true without statistical tests, as I mentioned before.

L213 - Where effective diffusivity is described?

Energy - probably after statstical analysis will be not significant differences between values of energy for 30,50 and 100W.

Statistical analysis and homogeneous groups as well as SD should be added to the Table 2 and Table 3.

L217 - eq. 15 not 12. Table 3, not table 3. results not result.

L225 - What does it mean that dE values are 18 and 17.68? The accuracy should be the same for each value.

L227-235 Without statistical analysis results do not stand for conclusions.

Reviewer 3 Report

Dear Editor,

I have read the manuscript entitled "Modelling and kinetic study of
novel and sustainable microwave-assisted dehydration of sugarcane
juice".

I would recommend the manuscript for extensive revision before it can
be published. My main objection is the lack of context. It is not made
clear why sugarcane juice can be an important health product. It is
mentioned that it has an effect on free radicals, perhaps this is the
suggested health benefit, but that relation is not made specific. In
addition, a comparison to alternative health products is absent, nor is
the need to investigate microwave processing instead of conventional
heating motivated. Could it possibly provide a better product? Then at
least there needs to be a comparison with conventional heating. This
comparison should include both criteria on product quality and energy
requirements.

In relation to energy in a processing context, it has to be noted that
microwave energy is relatively expensive. The benefits of applying a
microwave field must outweigh the energy cost for there to be a
business case. It is therefore important to develop an adequate energy
balance. This can be a complicated matter though, so I would advise the
authors to direct some attention to this matter. Microwave fields need
to be generated from electricity, which involves some losses, upon
which they are transmitted towards the process fluids that need to be
heated. This transmission is likely to coincide with field reflections
that constitute a reflection loss. Can this specific loss, or the
overall loss be quantified? If only through an approximate calorimetric
estimation this may already provide insight into the overall energy
expense, and hence may either support or discourage a business case.
Also in relation to energy, the Arrhenius-like equation expressed in
Eq. 14 has the microwave power "P" in the exponent expression, rather
than the usual "kT" expression. This is a significant deviation from a
regular physics description, and the authors must justify it by
elaborating their theoretical considerations behind this decision.

A final major concern that I have is the lack of a clearly defined
research question and methodology outline in the introduction of the
manuscript, and an answer to this question in the conclusions section.
In all this limits the manuscript to a display of craft, rather than
contextualizing it to real-world problems. Why, as a reader – either
from industry or academia – would I have an interest in reading it?

Reviewer 4 Report

General Comments

The manuscript entitled “Modelling and kinetic study of novel and sustainable microwave-assisted dehydration of sugarcane juice” is poorly written. Authors have to largely improve their work. In my opinion, the results presented do not allow such conclusions to be drawn. The Authors present the results for conventional concentrate only for solubility and color.

Some specific comments that may be useful while preparing the improved version of the manuscript :

L. 21, 24, 148. “100 (W)” should be replaced by “100 W”

P. 1 and 2. Introduction. In the manuscript, there is no specific aim of research.

L. 74, 148. “g water/(100 g.min)” should be replaced by “g water/100 g·min”

L. 79. How was determined Me?

L. 114. The results should be calculated by taking into account the dilution and expressed in μM trolox per g dry weight (dw). Antioxidant activity presented by Authors as the percentage of scavenging activity is incorrect. Therefore, presented results cannot be compared with results of other researchers.

L. 150. “100W and 30W” should be replaced by “100 W and 30 W” Between value and unit should be space.

L. 159. Figure 1c. Why the time shown on the chart is only up to 16 minutes. The concentration process of juice took longer. Please explain this. It is similar in the Figure 2 a.

L. 195. “watts” should be replaced by “W”

L. 205, 208,209,212. “(W)” should be replaced by “W”.

In general, the results should be presented as mean values and standard deviations (Tables 2 and 3). The Authors should better discuss the results of their studies and compare with the results of other researches. If the authors want to compare the concentration process of juice, they should use more methods of concentration juice. The manuscript requires intensive editing.

Round 2

Reviewer 1 Report

Most of the comments were taken into account. I have only a few minor corrections to the manuscript:

1. Table 1 - if you use in the text e.g. 10-2 instead of e, please use the same in the table

2. please, do not use abbreviations in the conclusion chapter (ER, MR).

Author Response

Table 1 - if you use in the text e.g. 10-2 instead of e, please use the same in the table

We have changed the format from “e” to “10-2”(Table:1) (line: 193)

please, do not use abbreviations in the conclusion chapter (ER, MR).

we have replaced the abbreviation with complete name in conclusion (line: 236).

Reviewer 2 Report

Once again is the same mistake in eq. 17 is "dried supernatent", but should be supernatant.

L133/134 Are still incorrect. I had written my suggestion: "Changes in color" and parameters WERE determined, not "was"!

Equation Em (microwave energy) is not numbered as eq. 17 (then, eq. should be numbered as eq. 18 and respectively).

L142 "to determine comparison?" or to determine differences between treatments?

L159, my previous sentence was: "L148 3.72 and 12.7 g - should be the same accuracy." Suggestion has been carried out (line :159).--> but is not corrected.

The title of Figure 1 is unclear, because in this figure is 3 figures, which are separated for a), b), c) (they are not described in the title).

Where is a reference to Figure 1c in the main text?

L172-173 are still unclear.

L174-175 (earlier "L166 - where are statistical data?" - your answer: The data is presented in tables (1,2 and 3) for respective parameters (lines-201, 226, 236) - in this lines are not statistical data. Where is a p-Value for an influence of microwave powers on effective diffusivity or energy consumption etc.???

L185-186 "the effect of various microwave powers on the constant of Midilli model" - what does it mean "on constant"?

L187-190 I still don't understand sense of these equations. They are not described on main text. What is "p"?

L198 - Name of Table 1 is still unchanged. Moreover, All the values should be the same accuracy!

L224 - Values of Specific energy consumption are similar I don't believe in added homogeneous groups in cases of Energy and Antioxidant activity. Why groups are not added in the case of Effective Diffusivity???

L229-230 "microwave-based heating exhibited better color properties
230 compared to conventional ones (Table. 3) i.e. ΔE & C* values are lower for microwave" This sentence is not true according to Table 3, because C* value for conventional heating is 3.85 and for microwave 4.07 - HIGHER! Moreover,  values of dE 18.01 and 17.68 are to similar - we cannot say that has better color properties, because is value is smaller about 0.33, which is unnoticeable by observer.

Author Response

1. Once again is the same mistake in eq. 17 is "dried supernatent", but should be supernatant.

The spelling has been corrected from “supernatant” to “supernatant” in equation 17 (line:128).

2. L133/134 Are still incorrect. I had written my suggestion: "Changes in color" and parameters WERE determined, not "was"!

The suggestion has been carried out (Line 133/134).

3. Equation Em (microwave energy) is not numbered as eq. 17 (then, eq. should be numbered as eq. 18 and respectively).

The equation for Em is marked as equation 17 and further changed the equation numbers accordingly  (line: 118).

4. L142 "to determine comparison?" or to determine differences between treatments?

The word comparison was replaced with difference in line: 142.

5. L159, my previous sentence was: "L148 3.72 and 12.7 g - should be the same accuracy." Suggestion has been carried out (line :159).--> but is not corrected.

The numbers after decimal were made homogenous throughout the manuscript (Line: 160).

6. The title of Figure 1 is unclear, because in this figure is 3 figures, which are separated for a), b), c) (they are not described in the title).

Title of figure 1 is modified to make it clear as per suggestion.

7. Where is a reference to Figure 1c in the main text?

It is discussed in lines 157-158.

8. L172-173 are still unclear.

These lines are modified to make clarity in the text for readers (lines: 169-170).

9. L174-175 (earlier "L166 - where are statistical data?" - your answer: The data is presented in tables (1,2 and 3) for respective parameters (lines-201, 226, 236) - in this lines are not statistical data. Where is a p-Value for an influence of microwave powers on effective diffusivity or energy consumption etc.???

We have added p-value for effective diffusivity, energy consumption and antioxidant activity in table 2 (line: 220).

10. L185-186 "the effect of various microwave powers on the constant of Midilli model" - what does it mean "on constant"?

This section has been removed from the text as it may confuse the readers.

11. L187-190 I still don't understand sense of these equations. They are not described on main text. What is "p"?

As stated earlier, this part is removed to avoid confusion.

12. L198 - Name of Table 1 is still unchanged. Moreover, All the values should be the same accuracy!

We have changed the name of table 1.  Besides, now all the values are of the same accuracy (line:193).

13. L224 - Values of Specific energy consumption are similar I don't believe in added homogeneous groups in cases of Energy and Antioxidant activity. Why groups are not added in the case of Effective Diffusivity???

The values of specific energy consumption are in converted into smaller units KJ instead of MJ to indicate the differences. Moreover, homogeneous groups to effective diffusivity were labeled in table 2 (line:220).

14. L229-230 "microwave-based heating exhibited better color properties
230 compared to conventional ones (Table. 3) i.e. ΔE & C* values are lower for microwave" This sentence is not true according to Table 3, because C* value for conventional heating is 3.85 and for microwave 4.07 - HIGHER! Moreover,  values of dE 18.01 and 17.68 are to similar - we cannot say that has better color properties, because is value is smaller about 0.33, which is unnoticeable by observer.

We agree with your suggestion, thus, text of these lines has been improved the text in the manuscript for ΔE & C* values (line: 226-230).

Reviewer 4 Report

General Comment

The manuscript entitled “Modelling and kinetic study of novel and sustainable microwave-assisted dehydration of sugarcane juice” is interesting. In my opinion, the article was improved to properly. However, I have minor comments that should be considered to make the manuscript suitable for publication.

L112 and 120. In my opinion “Kg” should be replaced by “kg”. International System of Units is abbreviated SI in all languages. Unit symbol of kilogram is "kg". This should be changed throughout the manuscript.

Author Response

L112 and 120. In my opinion “Kg” should be replaced by “kg”. International System of Units is abbreviated SI in all languages. Unit symbol of kilogram is "kg". This should be changed throughout the manuscript.

Suggestion has been carried out (lines 96,112,120, 203, 215 &220)

Round 3

Reviewer 2 Report

In my opinion this version of the manuscript is improved enough.